# Numerical Investigation of Scour Downstream of Diversion Barrage for Different Stilling Basins at Flood Discharge

Muhammad Waqas Zaffar [1,*], Ishtiaq Hassan [1], Umair Latif [2], Shah Jahan [3,4] and Zeeshan Ullah [4]

1   Department of Civil Engineering, Capital University of Science and Technology (CUST), Expressway, Kahuta Road Zone-V Sihala, Islamabad Capital Territory, Islamabad 44000, Pakistan; eishtiaq@cust.edu.pk
2   Communication and Work Department (C&W), Government of Punjab, Lahore 54000, Pakistan; umairlatif.cnw@gmail.com
3   Professional Engineer Construction Planning and Costing at DASU Hydropower Project, Kohistan 20100, Pakistan; shahjahan1002@gmail.com
4   NUST Institute of Civil Engineering (NICE), Department of Construction Engineering and Management (CE&M), National University of Science and Technology (NUST), Islamabad 44000, Pakistan; zeeshanullah.cem15@nit.nust.edu.pk
*   Correspondence: dce171001@cust.pk

**Abstract:** The hydraulic performance of stilling basins is affected by their size and geometry, which can be predicted by local scour. In 2008, based on a rigid bed study, the stilling basin of Taunsa barrage was remodeled, in which the old friction and baffle blocks were replaced with chute blocks and end sills. However, the study did not consider the effects of the remodeled basin on the erodible bed and only investigated hydraulic jumps. Therefore, this study developed FLOW-3D scour models for a designed flow of 24.28 m$^3$/s/m to investigate the flow field and local scouring downstream of old and remodeled basins. The results showed that as compared to Large Eddy Simulation (LES) and Standard K-ε models, the Renormalization Group (RNG-K-ε) model predicted the scour profiles with better accuracy, for which the coefficient of determination (R$^2$) reached 0.736, 0.823, and 0.747 for bays 33, 34, and 55, respectively. Downstream of the remodeled basin, the net change in sediment bed was 88%, 91%, and 95% in the LES, Standard, and RNG-K-ε models, respectively. However, downstream of the old basin, the net change in sediment bed reached 51%. Conclusively, based on the results, the study suggests investigating scour downstream of Taunsa Barrage using other discharges and sediment transport rate equations.

**Keywords:** barrage; retrogression; remodeling; stilling basin; scour; turbulence models

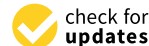



## 1. Introduction

Hydraulic structures such as weirs, sluice gates, and barrages are used to control and divert water into canals. Downstream of these hydraulic structures, a hydraulic jump (HJ) occurs, which dissipates and reduces the high kinetic energy of the upstream flow. HJ takes place when a supercritical flow changes into a subcritical flow. On the basis of Froude number (Fr), HJs are classified into four categories, such as weak ($1.7 < Fr_1 \leq 2.5$), oscillating ($2.5 < Fr_1 \leq 4.5$), steady ($4.5 < Fr_1 \leq 9$) and strong jump ($Fr_1 > 9$) [1].

The performance of any hydraulic structure can be affected if the HJs are not contained in the stilling basin [2]. Therefore, to stabilize the HJ and further reduce the energy, certain structural arrangements, such as baffles and friction blocks, are made within the stilling basin. Furthermore, after the rigid portion of the stilling basin, a flexible apron is provided downstream to avoid retrogression and scouring. Al-Mansori et al. conducted experiments to investigate the effects of different shapes of baffle blocks on energy dissipation [3]. Zaffar and Hassan investigated different flow characteristics in two different stilling basins using FLOW-3D. The results showed higher values of velocity, turbulent kinetic energy (TKE), and roller lengths in the remodelled stilling basin [4]. The results indicated that, as

compared to the United States Bureau of Reclamation (USBR), the new baffle reduced the length of HJ and increased the energy dissipation by 9.31%. Habibzadeh et al. [5] conducted experiments on the performance of baffle blocks for the submerged HJ. The results indicated two different flow regimes, such as Deflected Surface Jet (DSJ) and Reattaching Wall Jet (RWJ). The results also showed that DSJ was found to be more crucial in energy dissipation as it created reverse flow, which impinged on the free surface. Aydogdu et al. implemented Fluent 14.5 software to study the effect of different shapes of central sills in the expanding stilling basin [6]. Out of the studied sills, Type-3 sills produced more promising results in free surface profiles, velocity distribution, and roller length of HJ and TKEs. Ali and Kaleem [7] investigated different stilling basins of the Tuansa barrage, Pakistan, under various tailwater levels. The results indicated the remodeled barrage's stilling basin was not rational, which caused the launching of the stone apron and drifted the river toward the left bank. Chaudary and Sarwar [8] reviewed the designs of Tuansa barrage stilling basins and indicated that, as compared to the old stilling basin, the new stilling basin dissipated less energy. The study further showed that the tailwater rating curve was significant in keeping the HJ at the glacis. Chaudhry [9] used a one dimensional HEC-RAS model to study the free surface profiles in Taunsa Barrage's stilling basins. Tailwater, velocity, Froude number, and HJ location were the main investigated parameters. The results indicated that in the old stilling basin, HJ locations and velocities were within safe limits, while high flow depths were observed in the remodeled stilling basin, which affected different hydraulic parameters.

On the contrary, building a hydraulic structure on the pathway of environmental flows not only changes the channel bed but also affects the free surface profiles and flow patterns [10,11]. Consequently, these changes lead to scouring downstream of the hydraulic structure. The prediction of local scour downstream of the hydraulic structure is crucial because, if not properly calculated at the design stage, it can eventually undermine the foundation of the structure [12]. Therefore, investigation of local scouring downstream of the low and high-head hydraulic structures is an important research area because of the significant value of these structures. It is practically impossible to completely avoid the scouring; however, scour depth, upstream slope, and scour length are major intended parameters to minimize the risk of failure and have been investigated in many experimental and numerical studies. Yazdi et al. [13] investigated the effects of rectangular and trapezoidal geometries of the piano key weir on the local scour. After increasing the discharge and weir height for rectangular and trapezoidal models, the results showed an increase in the scour hole and its depth downstream of the rectangular model. Compared to the rectangular model, scour length was reduced by up to 11% in the trapezoidal model. Guan et al. [14] conducted experiments to investigate the turbulence structure and flow patterns in the scour hole downstream of the submerged weir. The results indicated that turbulence structures ahead of the large circulation zone were governing the depth and length of the scour hole. Balachandar et al. [15] investigated the effects of tailwater depths on the local scour downstream of a submerged sluice gate. The results showed that due to the oscillations of the fluid jet, velocity distribution was found to be complicated. During the digging phase, stream-wise and transverse velocities were directed towards the bed, while in the refilling stage, velocity vectors were directed to the free surface.

Conventionally, reduced-scale modeling is carried out to investigate the flow behavior of hydraulic structures. This modeling technique is expensive, time-consuming, and associated with scaling effects, i.e., models' and measurements' effects. Additionally, measuring devices also hinder the flow, and difficulties in displaying the terrain and concrete roughness reduce the accuracy of output variables. On the other hand, since the last thirty years, the use of numerical modeling has become prevalent to study the hydraulics of full-scale models such as FLOW-3D [16]. These models are efficient for investigating the complexity of spatial flow using turbulence models [17,18]. These turbulence models, i.e., Renormalization Group (RNG K-ε) [19], Large Eddy Simulation (LES) [20,21], and Standard k-ε [22,23] are able to test the internal structure of hydraulic jump, air entrainments, and

local scouring [24]. Since the development of FLOW-3D numerical models, several studies of HJ and other hydraulic parameters have been conducted, while little is known about scour modeling downstream of low-head hydraulic structures, i.e., barrages. However, a few studies on scour modeling using FLOW-3D have been witnessed, which are highlighted here. Aydin and Ulu [25] implemented FLOW-3D numerical models to investigate the energy dissipation and local scour downstream of the ogee spillway. The results showed that, compared to the other shapes, the triangular baffle block with a vertical face produced less scour. The results further showed that triangular baffle blocks with an end sill reduced the local scour by up to 90%. Le et al. investigated the local scour downstream of the box culvert using FLOW-3D [26]. The results showed scouring on the concave side, while deposition was noticed on the convex side of the downstream channel. The results also indicated that bed roughness/$d_{50}$ was the most sensitive parameter, which affected the scour depth and length. Tang and Puspasari [27] used FLOW-3D to investigate the local scour around three tandem piles. The results indicated that the downstream horseshoe vertex was producing scour, and the maximum scour was found around the first pile. Daneshfaraz et al. [28] investigated the effects of river harvesting material on the local scour around the bridge piers using FLOW-3D. The maximum scour was found around the downstream groups of piles. Shamohamadi and Mehboudi [29] investigated different parameters of scour in the confluence channel using FLOW-3D. The results showed that the ratio of discharge of the main channel to the secondary channel and the ratio of width of the main channel to the secondary channel were controlling the scour.

Barrages in Pakistan play a vital role in the economy, which was built about 50 to 100 years ago. With the passage of time, these barrages suffered due to hydraulic as well as structural deficiencies [30]. Similarly, Taunsa Barrage, Punjab, was also constructed about 81 years ago on the mighty River Indus for a design discharge capacity of 28,313 $m^3$/s. The stilling basin of the barrage was a modified form of the United States Bureau of Reclamation (USBR) Type-III basin, which consisted of impact friction and baffle blocks [4].

Soon after the barrage operation in 1958, multiple problems occurred on the barrage downstream, such as uprooting of the impact baffle blocks, damage to the basin's floor, lowering of tailwater levels, and bed retrogression. During 1959–1962, repairs were carried out to cater to these issues, but the problems remained persistent. To resolve these issues, the Punjab Government constituted committees of experts in 1966 and 1973, but no specific measures were taken, and the issues continued to aggravate. After so many partial repairs during the years 2004 to 2008, the stilling basin of the barrage was remodeled (USBR Type-II). Under the remodeling process, the impact friction and baffle blocks were replaced with chute blocks and end sills. For the remodeling of the stilling basin, a physical model study was conducted on the rigid bed [31], in which only the effects of tailwater levels on hydraulic jump were investigated. However, the model study did not provide the effects of the remodeled basin on the erodible bed. Even after the remodeling, during the years 2010–2014, probing data downstream of the barrage revealed that the block floor filter washed away in front of some bays. Additionally, the data further showed the sinking of the flexible apron downstream of the stilling basin. Based on the bibliographic analysis, it is found that only a few studies [7–9,32] have conducted hydraulic investigations on the stilling basins of the Tuansa barrage. However, the focus of this numerical study is to investigate the scour downstream of two different stilling basins of the investigated barrage. For the present study, FLOW-3D scour models are developed for the uncontrolled designed flow of 24.28 $m^3$/s/m. To further explore the scour patterns and bed retrogression, the study also implemented three different turbulence methods.

## 2. Materials and Methods

### 2.1. Study Area and Catastrophe of 2010 Flood

Taunsa barrage was constructed for a design discharge capacity of 28,313 $m^3$/s (24.28 $m^3$/s/m) and is located on the river Indus, as shown in Figure 1. The total width of the barrage between abutments is 1325 m, while 1177 m is the width for flow passage.

The old stilling basin of the barrage was jump-type and included friction and baffle blocks. These energy-dissipating devices were used to stabilize HJ and protect the stilling basin from scour and excessive retrogression [31]. However, multiple problems were observed after the barrage operation, and from 1958 to 2003, these problems were dealt with through partial repairs. From 2004 to 2008, the barrage stilling basin was remodeled, and old energy dissipation devices were replaced by chute blocks and end sills.

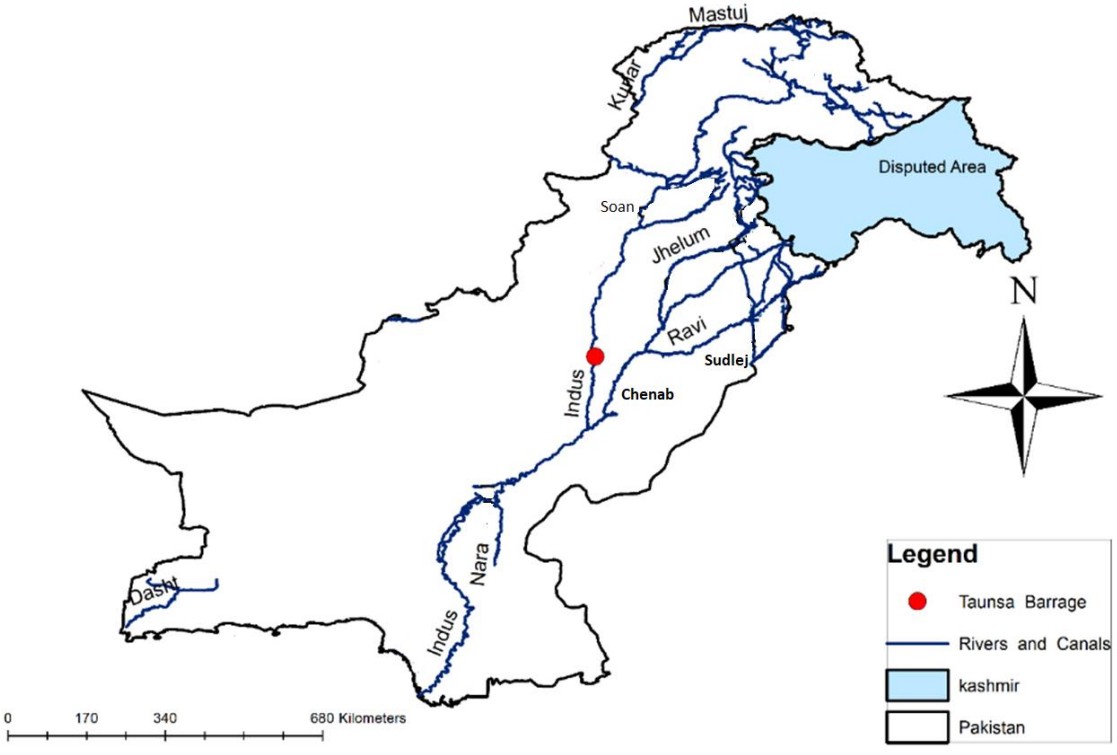

**Figure 1.** Location of the Taunsa barrage.

Floods in Pakistan occur due to heavy rainfall during the monsoon period and originate from the Bay of Bengal and Himalayan foothills. From 1950 to 2009, Pakistan suffered a total loss of US $20 billion, while the superfood of 2010 resulted in a total financial loss of US $10 billion. During the 2010 flood, Taunsa barrage could not pass the flood of 27,184 m$^3$/s (23.22 m$^3$/s/m), and its left marginal bund was breached, which led to a 7000 m$^3$/s discharge to the entire district of Muzaffargarh, Punjab. In total, during the 2010 flood, around 2000 people lost their lives, 17,553 villages were damaged, and a 160,000 km$^2$ area was affected. After the 2010 flood, sounding, and probing data downstream of the Taunsa barrage showed that in front of some bays, the filter block floor was washed away. Additionally, data revealed that the flexible apron downstream of the stilling basin was also sinking [33]. It is further reported that the remodeling of the stilling basin caused damage to the barrage downstream and raised the water level during the higher discharges.

For the present study, the geometry's details of the old (hereafter, called Type-A) and remodeled (hereafter, called Type-B) stilling basins of Taunsa Barrage used in the recent study of Zaffar and Hassan [4] are employed as shown in Figure 2a,b. However, downstream of the rigid bed, a FLOW-3D geometry window was used to construct the sediment scour beds.

### 2.2. Numerical Model Implementation

FLOW-3D scour models are developed to study the effects of Type-A and Type-B stilling basins. Before scour modeling, hydraulic models were operated to confirm their suitability for further analysis. Therefore, required meshing, initial and boundary conditions, turbulence models, and outputs were confirmed from the hydraulic stability of

different models. The proceeding sections provide the details of the adopted methodology, as shown in Figure 3.

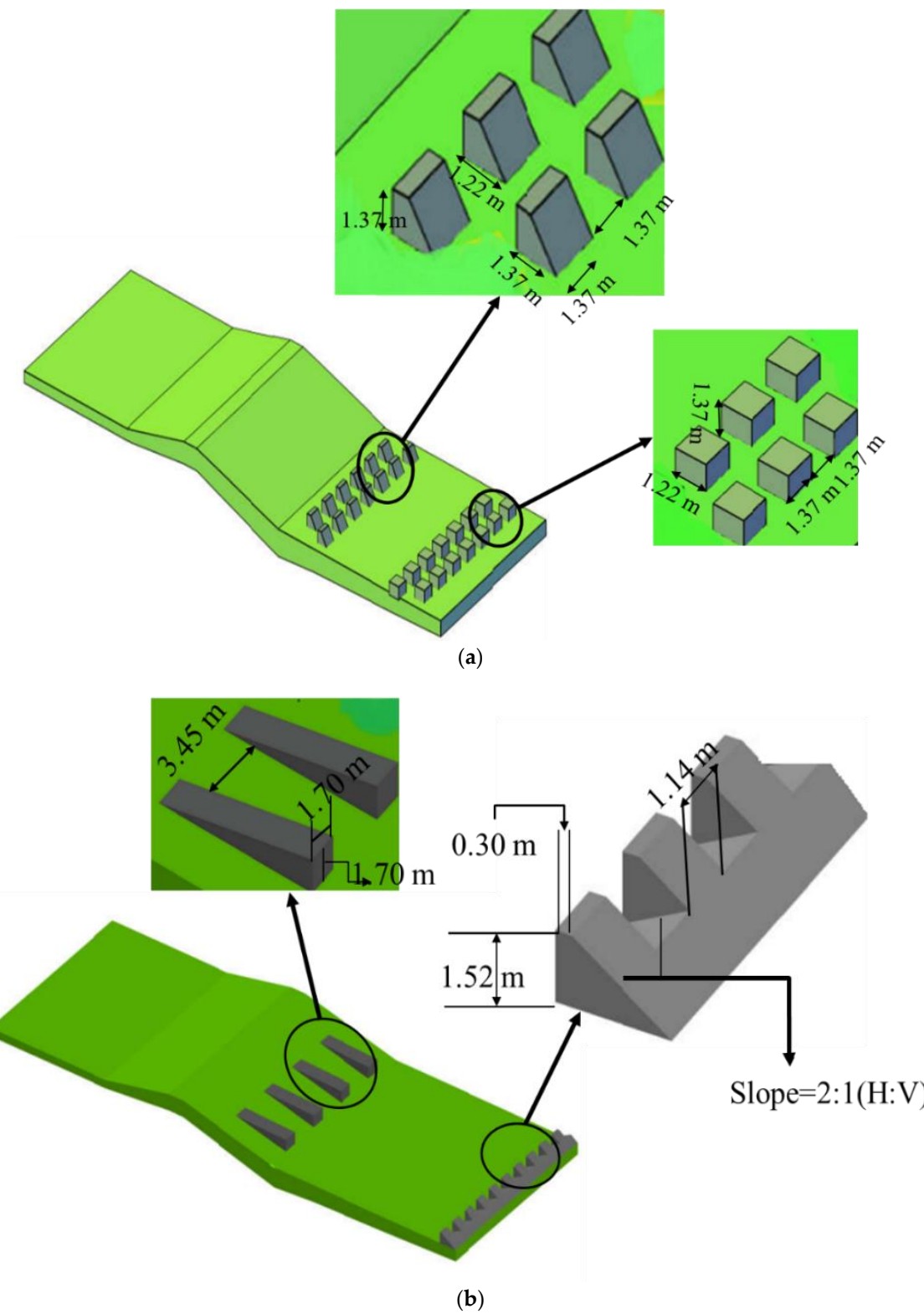

**Figure 2.** Barrage stilling basins, (**a**) Old (hereafter, called Type-A), (**b**) Remodeled (hereafter, called Type-B) (Zaffar and Hassan [4]).

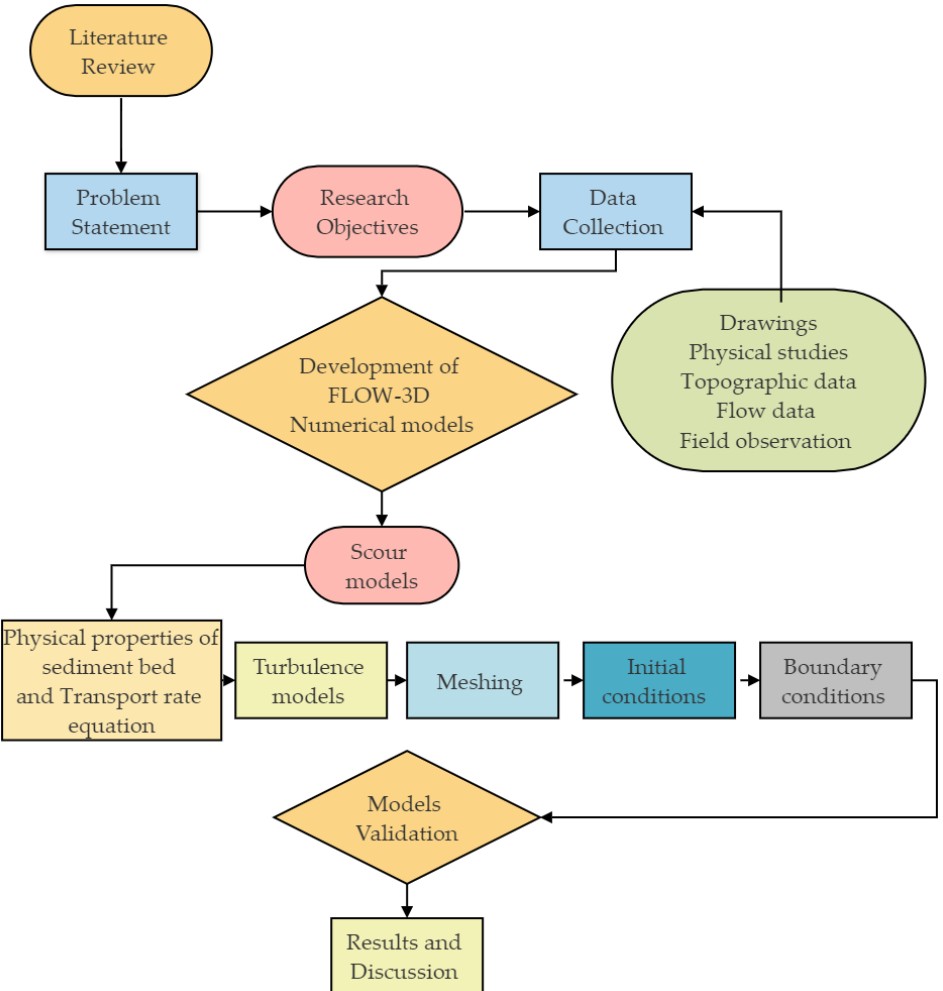

**Figure 3.** Flowchart for the present numerical models.

For three-dimensional (3D) flow analyses, FlOW-3D is one of the most potent numerical codes. The structured rectangular grids are used to resolve solid and flow domains. These grids store essential information on cells' faces/nodes. In FLOW-3D, the finite volume method (FVM) holds all the fluid's properties. For each computational cell, the Reynold Averaged Naiver Stokes (RANS) technique is used to discretize the Naiver Stokes and continuity equations. For a constant density $\rho$ of incompressible flow, FLOW-3D employs Equation (1) for the mass continuity equation.

$$\frac{\partial}{\partial_x}(\mathrm{u}A_x) + R\frac{\partial}{\partial_y}(\mathrm{v}A_y) + \frac{\partial}{\partial_z}(\mathrm{w}A_z) + \xi\frac{\partial_u A_x}{\mathrm{x}}(uA_x) = \frac{\mathrm{R}_{SOR}}{\rho} \tag{1}$$

where u, v, and w are the velocity components in the x, y, and z directions, Ax, Ay, and Az are the flow areas in the x, y, and z directions, $\mathrm{R_{SOR}}$ is the mass source, and R is the model coefficient.

The momentum equations for the x (2a), y (2b), and z (2c) directions are as follows:

$$\frac{\partial_u}{\partial_t} + \frac{1}{\mathrm{V_F}}\left[\mathrm{u}A_x\frac{\partial_u}{\partial_x} + \mathrm{v}A_y\frac{\partial_u}{\partial_y} + \mathrm{w}A_{zy}\frac{\partial_u}{\partial_z}\right] = \frac{1}{\rho}\frac{\partial_P}{\partial_x} + \mathrm{G}_x + \mathrm{f}_x \tag{2a}$$

$$\frac{\partial_v}{\partial_t} + \frac{1}{\mathrm{V_F}}\left[\mathrm{u}A_x\frac{\partial_u}{\partial_x} + \mathrm{v}A_y\frac{\partial_u}{\partial_y} + \mathrm{w}A_{zy}\frac{\partial_u}{\partial_z}\right] = \frac{1}{\rho}\frac{\partial_P}{\partial_y} + \mathrm{G}_y + \mathrm{f}_y \tag{2b}$$

$$\frac{\partial w}{\partial_t} + \frac{1}{V_F}\left[uA_x\frac{\partial u}{\partial_x} + vA_y\frac{\partial u}{\partial_y} + wA_{zy}\frac{\partial u}{\partial_z}\right] = \frac{1}{\rho}\frac{\partial P}{\partial_z} + G_z + f_z \tag{2c}$$

where u, v, and w are velocity components; $A_x$, $A_y$, and $A_z$ are the flow areas; $G_x$, $G_y$, and $G_z$ are body accelerations; $f_x$, $f_y$, and fz are viscous accelerations; $\rho$ is the fluid density; and P is the pressure.

To track the free surface, the following volume of fluid (VOF) Equation (3) is used. In the present study, only a single fluid (clear water) is considered.

$$\frac{\partial_F}{\partial_t} + \nabla \times (\bar{u}F) = 0 \tag{3}$$

where F is the fluid fraction and its value ranges from 0 to 1. If the value of F approaches 0, that indicates air in the cell, while F reaching 1 indicates water in the cell. However, F = 0.5 shows the free surface.

In computational fluid dynamics (CFD), one of the key aspects is turbulence modeling. RANS models create instability in the flow, for which turbulence models are used to find closure. These models add additional variables of turbulent viscosity and transport equations to solve the Reynolds stress term of Navier Stokes Equations (NSEs). In FLOW-3D, six turbulence models are available to solve closure issues; however, among these models, two equation turbulence models (turbulent kinetic energy, turbulent dissipation) such as Standard K-ε [20,21] and Renormalization Group (RNG K-ε) [34] are widely used in hydraulic and scour investigations. The formulation of both Standard K-ε and RNG K-ε is similar; however, they only differ in the derivation of model coefficients. At present, both Standard K-ε and RNG K-ε are used for scouring investigations. Additionally, to check the effects of eddies and recirculation in the scour zones, the study also employed the Large Eddy Simulation (LES) turbulence scheme [35].

### 2.3. Sediment Scour Model

In FLOW-3D, the sediment transport model is described by bed and suspended transport loads. The bed load transport phase includes advection, erosion, settlement, and deposition of the sediment particles. For scour modeling, some of the most crucial parameters, such as critical shield number, bed load transport rate equation, maximum packing fraction, and bed shear stress, are defined as follows.

To remove the sediment particle from the bed, a critical shield number is required, which depends on the sediment size, density, and forces acting on the particle. The following Equation (4) is used to compute the critical shield number. In FLOW-3D, critical shield numbers can be calculated from the Soulsby–Whitehouse equation and can also be selected by the user. Presently, user defined values are used.

$$\theta_{cr,i} = \frac{\tau_{cr,i}}{gd_i\,(\rho_i - \rho_f)} \tag{4}$$

where g is acceleration due to gravity, $\tau_{cr}$ is the critical shear stress, $d_i$ is the diameter of sediment, $\rho_i$ and $\rho_f$ and are the sediment and fluid densities.

For the flat riverbed assumed in the present study, Equation (5) is used for critical shear stress ($\tau_{cr}$).

$$\tau_{cr} = \rho g\,(s - 1)d_{50}\theta_{cr} \tag{5}$$

where $\tau_{cr}$, s, $d_{50}$, and $\theta_{cr}$ are the critical shear stress, specific density, mean diameter, and critical Shields number for the sediment particle, respectively.

In the present model, because of the non-cohesive bed [26–28,36,37], Van Rijn Equation (6) is implemented to calculate the dimensionless bed load transfer rate.

$$\Phi_i = \beta_{VR,i}\,\text{d}^*,\,i - 0.3\,\left(\frac{\theta i}{\theta cr, i} - 1.0\right)2.1\,c_{b,I} \tag{6}$$

$\beta_{VR,i}$ is the bed load coefficient, which has a default value of 0.053, $C_{b,I}$ is the fractional volume of i species in a packed sediment bed, and ($\theta_{cr,i}$) is the critical shield parameter. The physical properties of the sediment-packed bed and other essential parameters that affect the scouring process are provided in Table 1.

**Table 1.** Physical properties of the erodible bed and essential parameters governing scour.

| Sr. No. | Sediment Characteristics | Modelling Value |
|:---:|:---:|:---:|
| 1 | Bed Load species | Fine sand |
| 2 | Species diameters ($d_{50}$) | 0.0002 m |
| 3 | Sediment density | 1692 kg/m$^3$ |
| 4 | Critical Shield number | 0.05 |
| 5 | Entrainment coefficient | 0.018 |
| 6 | Bed load coefficient | 0.053 |
| 7 | Angle of repose (Degree) | 32° |

### 2.4. Model Geometry and Meshing

Solid geometries of different models were prepared in AutoCAD, and their stereolithography (stl.) files were imported in FLOW-3D. The gate and sediment packed bed (scour bed) were created by the FLOW-3D geometry window. It is important to mention that the assigned sediment-packed bed was specified as 100% sand. Figure 4 shows the geometrical setup for the different studied models.

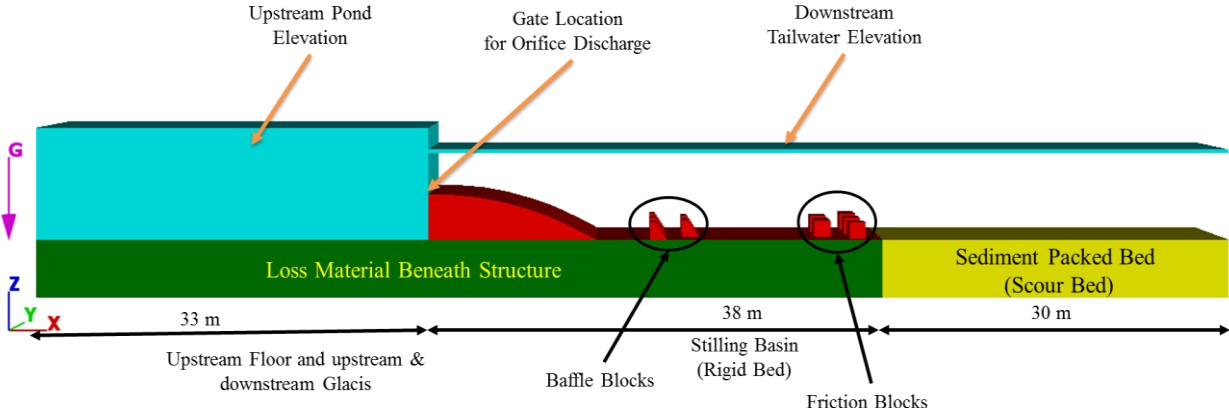

**Figure 4.** Three-dimensional setup for a rigid and erodible bed.

The entire solid and fluid domains were resolved by two structured mesh blocks. The first mesh blocks initiated from upstream ($X_{min}$ = 15 m), which ended downstream ($X_{max}$ = 71 m). However, the second mesh block started at $X_{max}$ = 71 m and ended at $X_{max}$ = 100 m, which also included 30 m of erodible bed as shown in Figure 4. It is worth mentioning that in the first mesh block, HJ and flow characteristics were closely monitored, while the second block was used to investigate the flow fields and scour behaviors. The details of the mesh blocks, cell size, and mesh quality indicators are presented in Tables 2 and 3, respectively.

**Table 2.** Cell characteristics in different mesh blocks.

| Mesh Blocks/ Cell Characteristics | Mesh Block-1 | Mesh Block-2 |
|:---:|:---:|:---:|
| Cell size | $\Delta x$ (m), $\Delta y$ (m), $\Delta z$ (m) 0.15, 0.20, 0.15 | $\Delta x$ (m), $\Delta y$ (m), $\Delta z$ (m) 0.25, 0.25, 0.25 |
| Total Cells | 1,492,000 | 222,720 |

**Table 3.** Mesh quality indicators in different mesh blocks.

| Mesh Block | Number of Cells | Maximum Adjacent Ratio | | | Maximum Aspect Ratio | | |
|---|---|---|---|---|---|---|---|
| | | X | Y | Z | X-Y | Y-Z | Z-X |
| Block-1 | X = 373, Y = 40, Z = 100 | 1.00 | 1.00 | 1.00 | 1.30 | 1.30 | 1.0 |
| Block-2 | X = 116, Y = 32, Z = 60 | 1.00 | 1.00 | 1.00 | 1.01 | 1.01 | 1.00 |

### 2.5. Initial and Boundary Conditions

Table 4 illustrates that both upstream and downstream boundaries of mesh block-1 were set as pressure (P), while lateral sides ($Y_{min}$, $Y_{max}$) and floor ($Z_{min}$) were set to wall (W) boundaries, which expressed no-slip and zero tangential and normal velocities ($u = v = w = 0$) to the walls, whereas u, v, and w are velocities in x, y, and z directions, respectively. In the case of mesh block-2, upstream and downstream boundaries were set as symmetry (S) and pressure (P), respectively, while lateral ($Y_{min}$, $Y_{max}$) and floor ($Z_{min}$) boundaries were set like in mesh block-1. For all variables except pressure (which was set to zero), the upper boundaries ($Z_{max}$) for both mesh blocks were set to atmospheric pressure, which allowed the fluid to null von Neumann. The initial conditions for the models' operation are provided in Table 5.

**Table 4.** Boundary conditions governing the simulations.

| Mesh Blocks | Upstream ($X_{min}$) | Downstream ($X_{max}$) | Lateral ($Y_{min}$) ($Y_{max}$) | | Top ($Z_{max}$) | Bottom ($Z_{min}$) |
|---|---|---|---|---|---|---|
| Block-1 | Pressure | Pressure | Wall | Wall | Atmospheric Pressure | Wall |
| Block-2 | Symmetry | Pressure | Wall | Wall | Atmospheric Pressure | Wall |

**Table 5.** Initial hydraulic conditions for the models' operation.

| Discharge (m³/s/m) | Pond Level (m) | Tailwater Elevation (m) | Barrage Operation | Turbulence Models |
|---|---|---|---|---|
| 24.28 | 135.93 | 134 | Free designed Flow (Uncontrolled Free condition) | 1-Standard K-ε 2-RNG K-ε 3-LES |

### 2.6. Operation of Numerical Models

In actuality, over a certain period, flows passing through the barrage differ from one bay to another. Under such conditions, the barrage goes through four different flow conditions: controlled free (CF), controlled submerged (CS), uncontrolled free (UCF), and uncontrolled submerged (UCS), and under these situations, different patterns of local scour and retrogression occur downstream of hydraulic structures. Following Figure 5a, the present uncontrolled free (UCF) condition was assessed for the scour downstream of different stilling basins. For developing uncontrolled free flow (UCF) conditions, models were operated for $H/H_d = 0.998$ (Savage and Johnson [38]; Johnson and Savage [39]; Gadge et al. [40]; Ghosh et al. [41]; Bhosekar et al. [42]) on pond and tailwater levels of 135.93 m and 134 m, respectively, whereas H (135.93 m) and $H_d$ (136.24 m) are effective and deigned heads. For the calculation of flow, the following Formula (7) was implemented.

$$Q = \frac{2}{3} * C_d * B * H_0^{3/2} * \sqrt{2g}$$ (7)

where Q (m³/s), $C_d$, B (m), and $H_0$ (m) are discharge, coefficient of discharge, bay width, and total energy over the crest ($H_0 = H + U^2/2g$; whereas U is the approach velocity),

respectively, while g (m/s$^2$) is the gravitational acceleration and P is the weir height. The definition sketch of uncontrolled free (UCF) flow conditions is provided in Figure 5b.

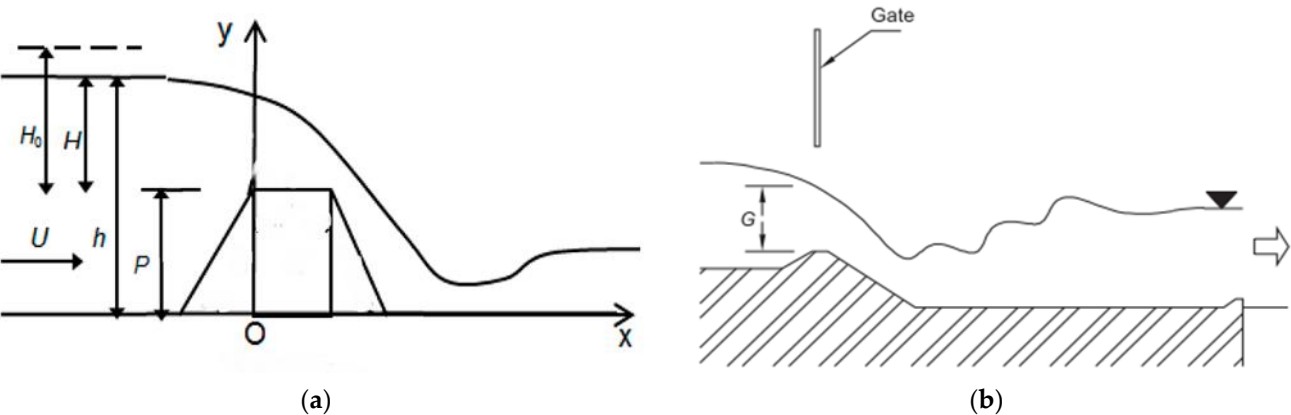

(**a**)   (**b**)

**Figure 5.** (**a**) Schematic diagram of uncontrolled free (UCF) flow conditions for assessment of scour and bed retrogression, (**b**) Definition sketch of uncontrolled free (CF) flow.

## 3. Results

For stability and convergence, the time step in each iteration was controlled by the courant number. It is worth mentioning here that, in full-scale numerical models on free flows such as 24.38 m$^3$/s/m, solution convergence and steady state of the models can only be achieved by hydraulic parameters such as volume flow rates and mass-averaged fluid kinetic energy (MFKE) at the inlet and outlet boundaries. Therefore, solutions for the scouring of uncontrolled free flow (UCF) were carried out for Ts = 300 s, where Ts is the end simulation time. In a Type-A stilling basin using the LES turbulence scheme, the model showed hydraulic stability at different intervals of time such as 81 s, 111 s, 141 s, 171 s, 200 s, 231 s, 261 s, and 291 s, while even at the finish time of T = 300 s, no hydraulic stability was noticed in standard K-ε and RNG K-ε models. At the simulation end in the Type-A stilling basin, the values of MFKEs were 8.41 m$^2$/s$^2$, 8.56 m$^2$/s$^2$, and 8.42 m$^2$/s$^2$ in the LES, Standard K-ε and RNG K-ε models, respectively. On the other hand, upon use of the LES model, the flow hydraulic stability in the Type-B stilling basin was found to be achieved at several time intervals, such as 68 s, 93 s, 118 s, 143 s, 168 s, 193 s, 218 s, and 243 s. However, using Standard K-ε and RNG K-ε models, the initial hydraulic stability in the Type-B stilling basin was achieved at T = 121 s and T = 222 s, respectively. The final values of MFKEs in the Type-B stilling basin were 8.67 m$^2$/s$^2$, 8.95 m$^2$/s$^2$, and 8.72 m$^2$/s$^2$ in the LES, Standard k-ε and RNG k-ε models, respectively. Convincingly, simulation finish time (T = 300 s) was found to be acceptable for such a value of discharge because, at these flow rates, free surface profiles are always found to be fluctuating. On the contrary, scour did not achieve the stability condition, and the sediment bed continued to erode even after the actual averaged simulation time of Ta = 77 h. Additionally, the computed free surface and bed profiles downstream of the investigated stilling basins were found to be changing because of the gradual bed's erosion. As the present study compares the effects of dissimilar stilling basins on the downstream riverbed, the defined simulation time at which the maximum riverbed exposure was focused was chosen to highlight the results. At T = 300 s, downstream of Type-B stilling basin, net height change in sediment bed was reached at 95% using the RNG K-ε model; thereby, the modeling time was found to be sufficient to compare results of flow field and scour for Type-A and B stilling basins.

### 3.1. Models' Validation

The present scour models were developed downstream of the Taunsa barrage in stilling basins. The Type-A stilling basin was originally designed in 1958 and remained functional until 2003. In 2008, the stilling basin of the barrage was remodeled (Type-B), and after its remodeling, during years 2010 and 2013–2014, severe damages on its downstream

were reported [7–9,32]. Therefore, the applicability of the present models to observe the scour and their validation are confirmed by the field data. Following Figure 6, field data represents the bed profiles downstream of the Type-B stilling basin; therefore, the results of the bed profiles downstream of the Type-B stilling basin are used for validation purposes. During the flood, flow with extremely high velocity passes through the barrage structure, and it is impossible to carry out a physical inspection of scour until the flows in the rivers are low [41]. To examine the bed profiles, depth-measuring probes are usually dropped from the boats. Hence, probing data from 2015 was digitized, and approximate bed profiles from the centerline of bays 33, 34, and 55 are drawn in AutoCAD. Using three different turbulence models, scour profiles were compared with the probing data of the above-mentioned bays.

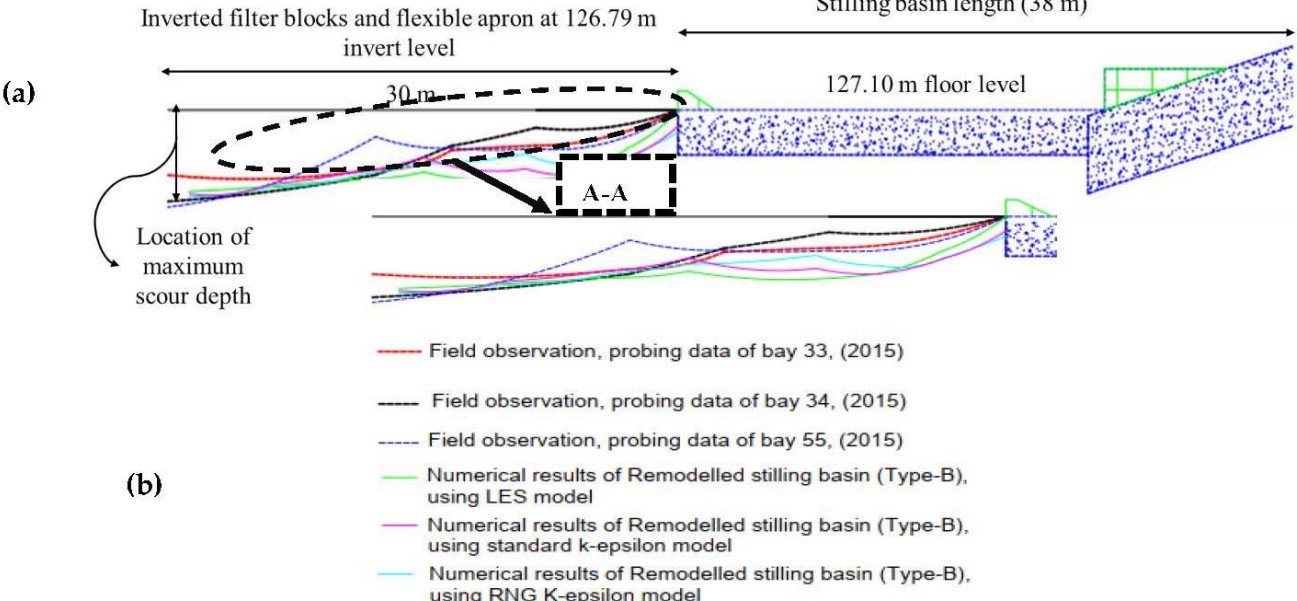

**Figure 6.** Comparison of scour profiles, (**a**) Type-B stilling basin and scour profiles (**b**) Section A-A for observed and predicted scour profiles.

The bed profiles from the present scour models were obtained at a designed discharge of 24.28 m³/s/m and then superimposed in the AutoCAD files for comparison. At the finished time, the sediment bed downstream of Type-B stilling was scoured up to 88, 91, and 95% in LES, Standard K-$\varepsilon$ and RNG K-$\varepsilon$, respectively. Figure 6 compares the modelled scour profiles downstream of the Type-B stilling basin with the observed data of bays 33, 34, and 55. On comparison with observed data from bay 33, the results showed that the LES and Standard K-$\varepsilon$ models overestimated the scour for the initial 12 m length of sediment bed, while the scour profile in the RNG K-$\varepsilon$ model was found to be close to the observed data. However, at the end of the sediment bed, the profiles produced by LES and Standard K-$\varepsilon$ followed the observed data. After comparing with the observed data of bays 34 and 55, the result of the scour profile using the RNG K-$\varepsilon$ model showed reasonable accuracy, and the trends of the present model also showed agreement with Ghosh et al. [41] and Kim et al. [43], as shown in Figure 6b. Based on the results, it is noticed that the RNG K-$\varepsilon$ model has predicted the scour profiles better than the LES and Standard K-$\varepsilon$ models. Upon use of the RNG K-$\varepsilon$ turbulence model, the analysis further revealed that out of the compared scour profiles, the results of bay 34 were found to be in good agreement with the field data. Convincingly, it can be said that the present scour models predicted the bed profiles with acceptable accuracy, which allowed us to analyze the flow fields and scour patterns. After applying regression analysis, the prediction of the present numerical models was also assessed by the coefficient of determination ($R^2$) [24,28]. The scour profiles downstream of the prototype bays 33, 34, and 55 were compared with all three turbulence models, i.e., the

LES, Standard K-ε, and RNG K-ε models, as presented in Figure 7. Following Figure 7a–c indicated the scour prediction by LES model, and the results indicated 0.523, 0.402, and 0.488 values of $R^2$ for bays 33, 34, and 55, respectively. Upon use of the Standard K-ε model, the predicted scour profiles for bays 33, 34, and 55 are shown in Figure 7d–f, respectively.

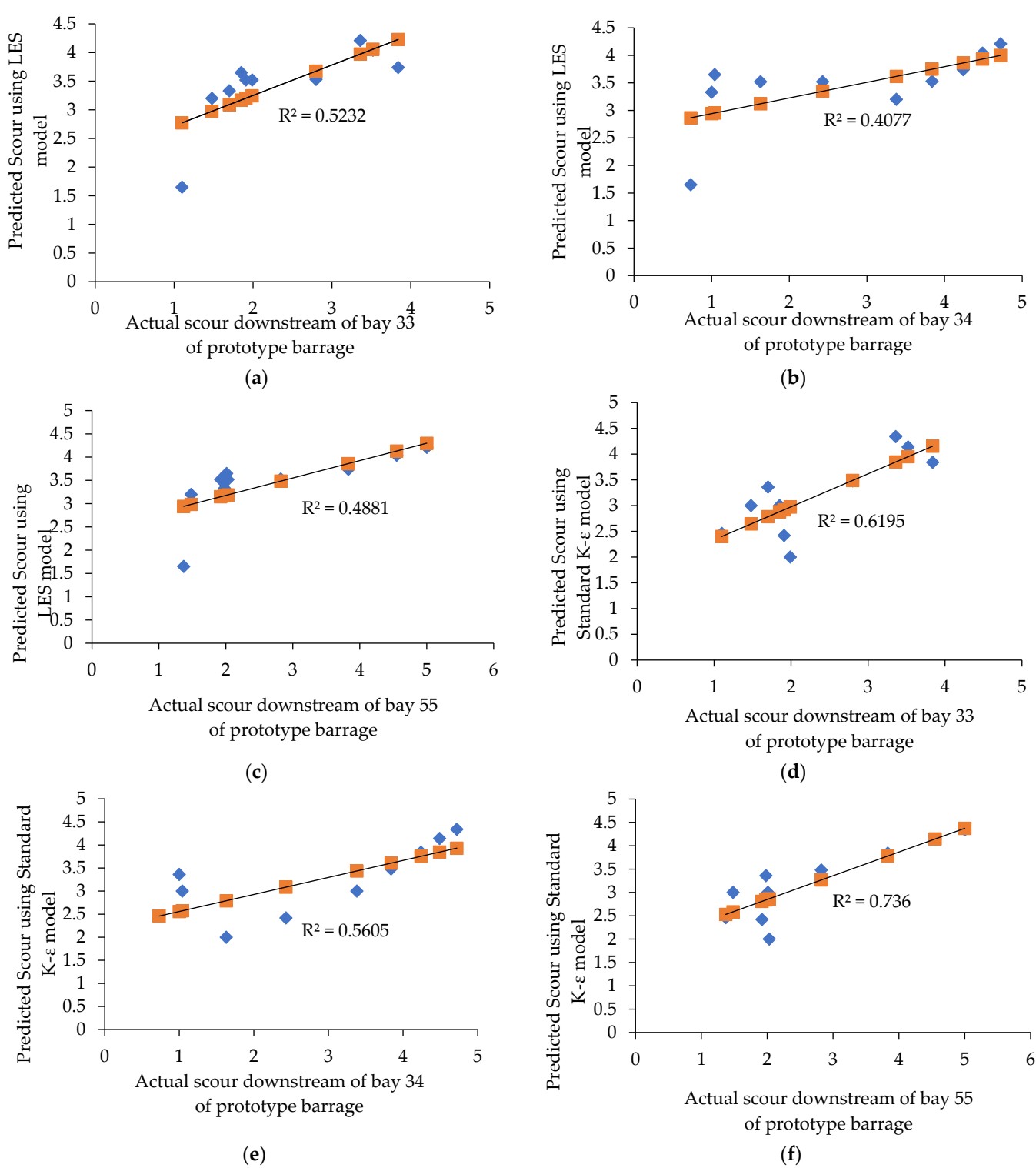

**Figure 7.** *Cont.*

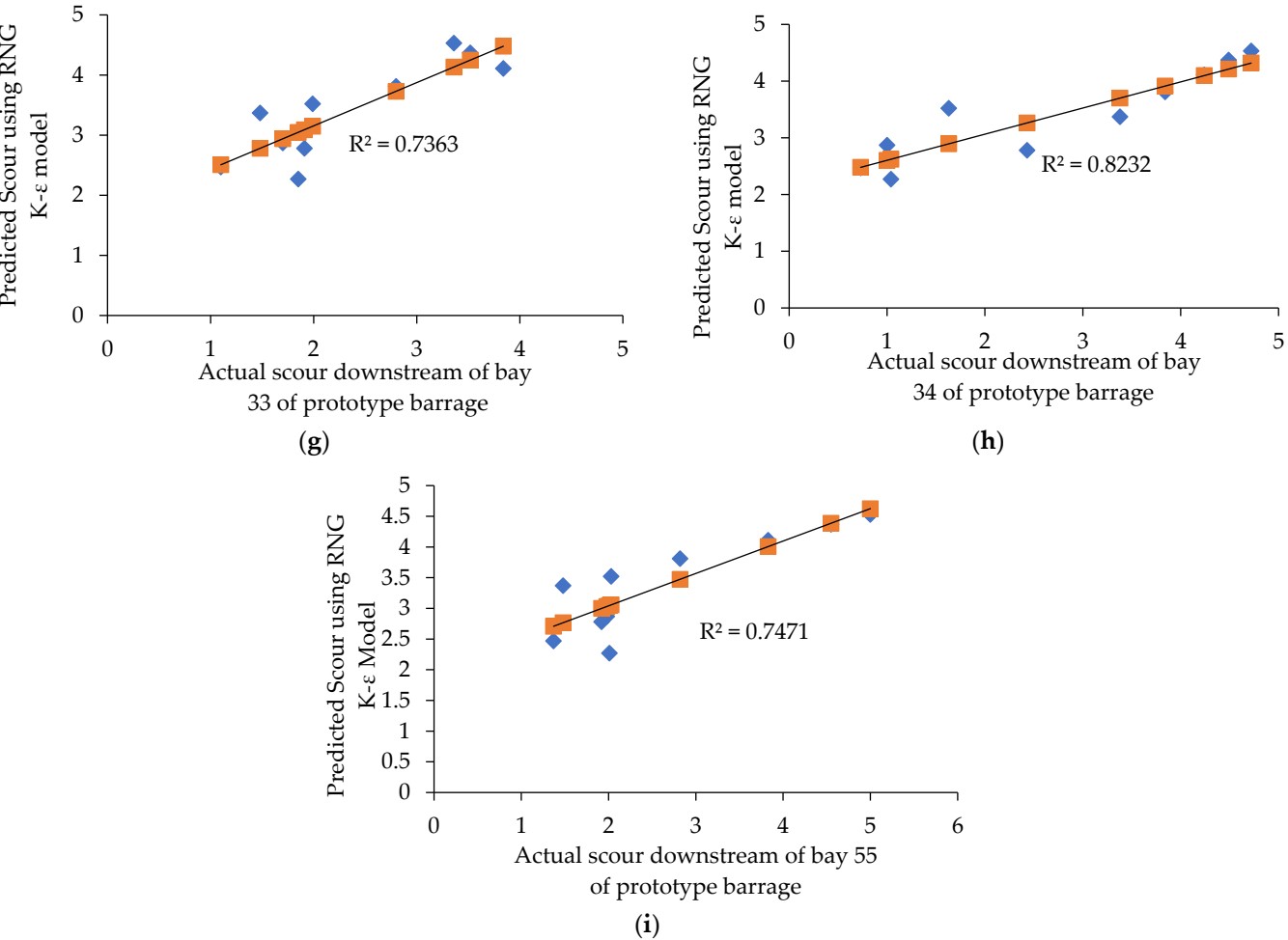

**Figure 7.** Comparison of observed scour profiles downstream of prototype bays with numerical models using different turbulence schemes, LES Model (**a**–**c**); Standard K-ε (**d**–**f**); RNG K-ε (**g**–**i**).

The results indicated 0.619, 0.560, and 0.736 values of $R^2$ for bays 33, 34, and 55, respectively. Upon use of standard K-ε, the prediction results of bay 55 were found to be close to the prototype scour profile, as shown in Figure 7f. Upon use of the RNG K-ε model, the prediction of the present models was found to agree with the prototype profiles. In the RNG K-ε model, the value of $R^2$ reached 0.736, 0.823, and 0.747 for bays 33, 34, and 55, as shown in Figure 7g–i, respectively. Based on the results, it can be said that the RNG K-ε model has predicted the scour profiles better than the LES and Standard K-ε models. The results further revealed that out of the compared scour profiles downstream of the prototype barrage's bays, using the RNG K-ε turbulence model, the results of bay 34 were found to be in good agreement with the field data. Convincingly, it can be said that the present scour models predicted the bed profiles with acceptable accuracy, which allowed us to analyze the flow fields and scour patterns.

### 3.2. Flow Field on Scour Bed

To understand the phenomenon of scour and bed retrogression downstream of Type-A and Type-B stilling basins, this Section 3 describes the flow field on the sediment bed using different turbulence models. All the studied models were run at 24.28 m³/s/m, and the results of the flow field are represented from the centerline of the bay. Figure 8a indicates the flow field on the scoured bed downstream of the Type-B stilling basin using the LES turbulence model. On the sediment bed, the LES turbulence model developed an irregular flow field and streamline pattern, as shown in Figure 8a. Similarly, the free surface profile over the bed was also found to be turbulent. In the Type-B stilling basin, at the start of the

sediment bed, the end-sill deflected the flow in an upward direction, which showed high values of velocity vectors, while minimum values of velocity vectors were noticed near the scoured bed, as shown in Figure 8b. Moreover, near the scoured bed, large recirculation and wake zones were noticed, which eroded the bed, and this gradual process of erosion caused a large scour hole beneath the wake zone. Figure 8c,d show the velocity contours and velocity distribution downstream of the Type-A stilling basin using the LES model. At the start of the sediment bed, higher velocity values were noticed in the middle and upper fluid regions, which declined as the fluid moved toward the end of the scoured bed.

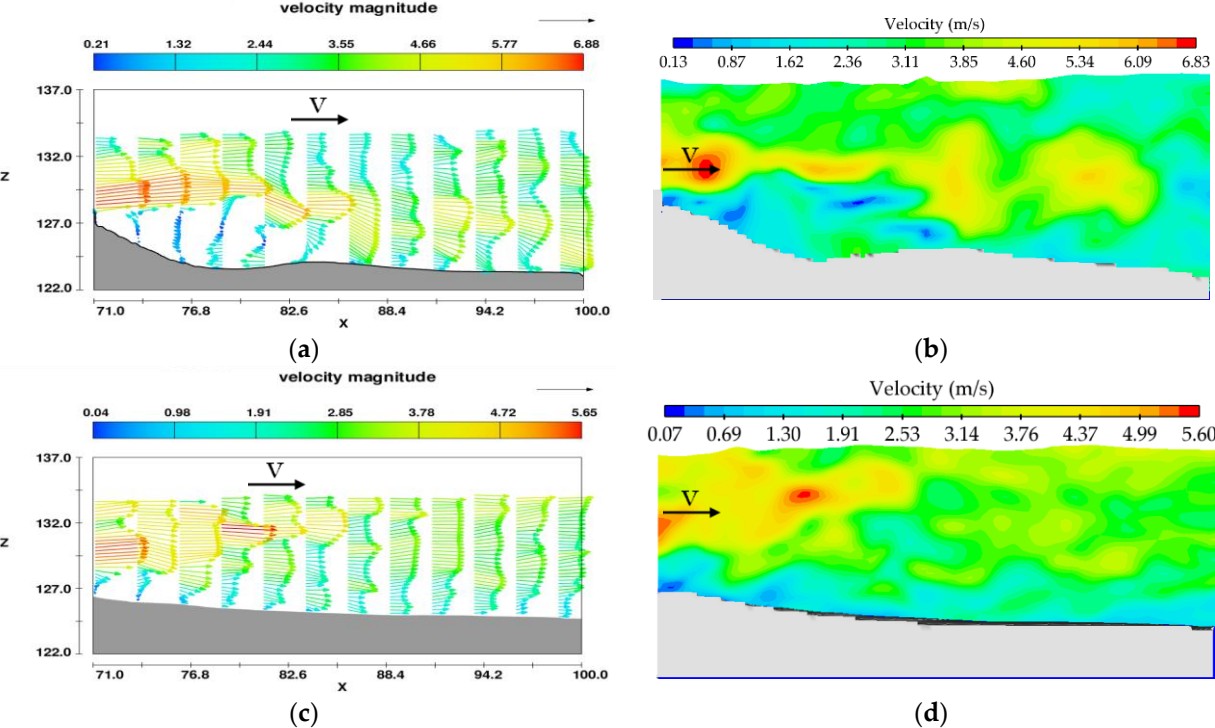

**Figure 8.** Flow field illustrating velocity vector and distribution on the scoured bed using the LES turbulence scheme. (**a**,**b**) Type-B stilling basin and (**c**,**d**) Type-A stilling basin.

The following Figure 9 shows the flow pattern and vectors' behavior downstream of Type-A and B basins using the Standard K-ε turbulence model. For both basins, as compared to the LES model, Standard K-ε displayed different patterns of flow fields on the sediment bed. In the Type-B basin, like in the LES model, maximum values of velocity were found in the middle fluid zone, which was deflected by the end-sill, as shown in Figure 9a. Following Figure 9a, results further showed that as the distance from the start of the sediment bed increased, the position of maximum velocity vectors in the streamlines was lowered. In the Type-B stilling basin, like the LES model, the Standard K-ε models also showed large recirculation and wake zones just after the rigid bed, as shown in Figure 9a,b. On the contrary, upon use of standard K-ε, only forward velocity profiles were noticed downstream of the Type-A stilling basin, which leveled off on the downstream side as shown in Figure 9c. Figure 9c also shows that at the start of the sediment bed, the intensity of slow-moving velocity vectors and the wake zone near the bed were at their minimum, which increased as the flow traveled downstream. Additionally, Figure 9d also shows that as the distance from the start of the sediment bed increased, the depth of slow-moving velocity vectors also increased.

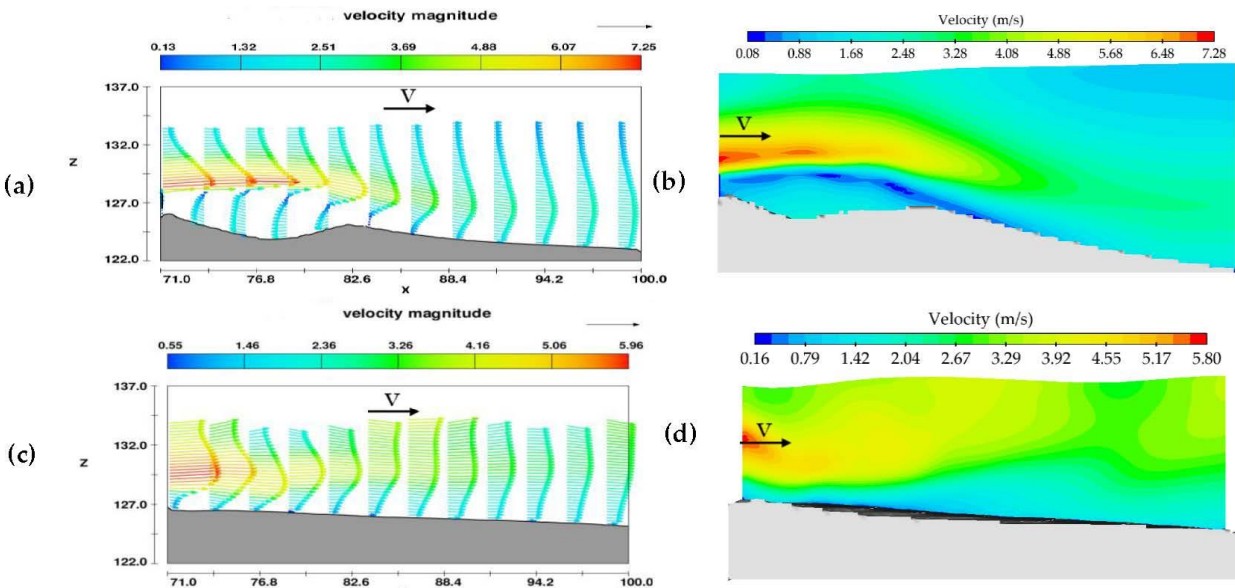

**Figure 9.** Flow field illustrating velocity vector and distribution on the scoured bed using the standard K-ε turbulence model. (**a**,**b**) Type-B stilling basin and (**c**,**d**) Type-A stilling basin.

Above, Figure 10 shows the flow field on the sediment bed using the RNG K-ε model. In both the studied basins, the RNG K-ε turbulence model showed the same flow behaviors and pattern of velocity vectors as the Standard K-ε model. However, in the RNG K-ε model, forward velocity profiles leveled off earlier, and compared to the LES and Standard K-ε models, the net change of bed was higher in the RNG K-ε. Figure 10a,b show that after the large recirculation zone, high velocity vectors were moved towards the bed, which caused bed retrogression downstream of the Type-B stilling basin. It is important to mention that below the large recirculation zone, a large scour hole was developed. On the other hand, upon use of the RNG K-ε model, forward velocity vectors were noticed downstream of the Type-A basin, and maximum velocity values were noticed in the central region, which shifted to the upper section of the fluid as the flow moved on downstream. Throughout the scoured bed, the lower zone of fluid was comprised of low-moving flow, which caused bed retrogression as shown in Figure 10c,d. However, at the end of the eroded bed, the depth of slow-moving flow increased, which also increased the erosion in those regions, as can be seen in Figure 10d.

*3.3. Scouring Analysis and Bed Profiles*

Figure 11a shows the contour plots of scour in a Type-A stilling basin using the LES model. Using the LES model, 48% of the bed was retrogressed at the end locations.

Figure 11b illustrates the scour pattern in a Type-A stilling basin using the Standard K-ε model. As compared to the LES model, a less scoured bed was observed in the Standard K-ε model, which indicated less scouring on the upstream side of the sediment bed. From a rigid bed to X = 2 m, a 0.2 m bed was retrogressed, while in the LES model, the net change in bed was up to 0.7 m. Notably, at the end of the sediment bed, Standard K-ε showed a lower change in bed height, which indicated a 43% reduction in the bed height. Figure 11c displays the contours of scour on the sediment bed downstream of the Type-A stilling basin using the RNG K-ε model. As compared to the Standard K-ε model, the initial length of the sediment bed up to X = 5 m was more scoured, while the final scoured profile of the bed was found to be identical to the Standard K-ε model. The net change of sediment bed downstream of Type-A stilling using the RNG K-ε model was 51%. As compared to Standard K-ε and RNG K-ε models, the overall length of bed change (from X = 83 to 95 m) was greater in the LES model, as can be seen in Figure 11a. Figure 12 shows the contours of the scoured bed downstream of the Type-B stilling basin using different turbulence models.

Figure 12a shows the contours of the scoured bed by the LES model. The results showed a scour hole just downstream of the end sill, and its length reached 15 m. The upstream side of the scour hole was eroded rapidly as compared to the downstream side. The width of the scour hole was found through the width of the model bay. The maximum depth of the scour hole was reached at 3.65 m, and after the scour hole, the bed retrogressed more rapidly up to the end of the sediment bed. The total change in the scoured bed downstream of the Type-B stilling basin reached 76% and 88% in the scoured hole and end of the sediment bed, respectively. Figure 12b shows the scoured bed downstream of the Type-B stilling basin using the Standard K-ε model. As compared to LES, the length and depth of scour were found to be less in the Standard K-ε model. The maximum depth of the scour hole was reached at 3.35 m, while the maximum depth at the end of the sediment bed was 4.34 m. Conclusively, the net change of bed in the scour hole and end of stilling, using the Standard K-ε model, reached 70% and 91%, respectively.

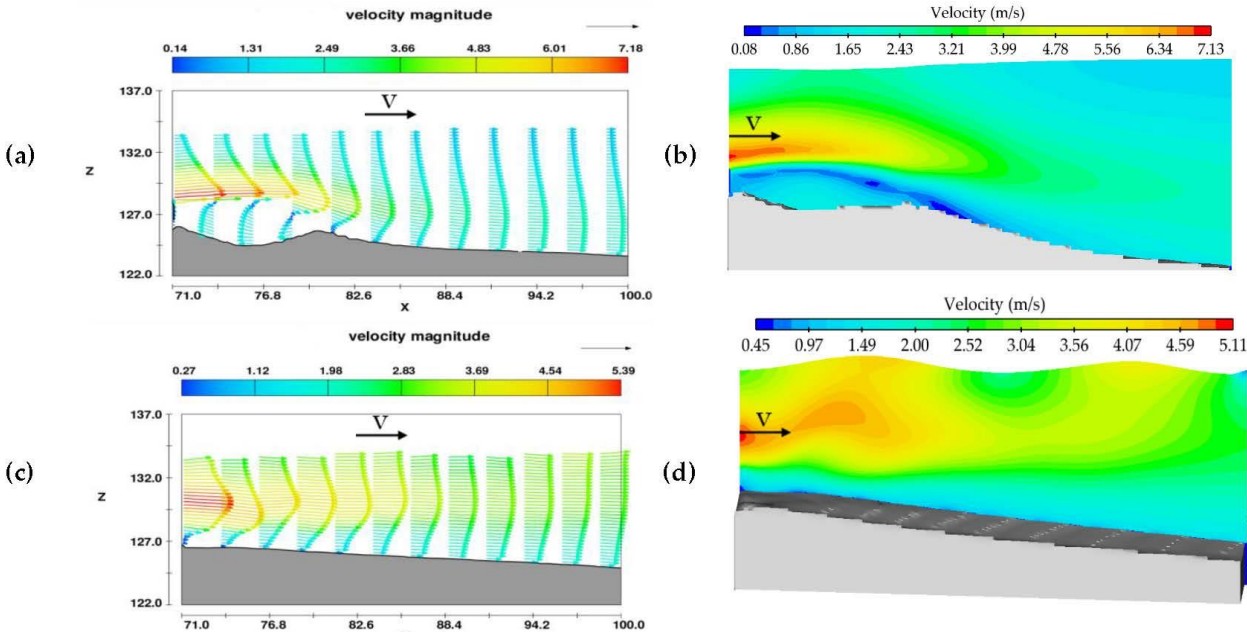

**Figure 10.** Flow field illustrating velocity vector and distribution on the scoured bed using the RNG K-ε model turbulence model. (**a**,**b**) Type-B stilling basin and (**c**,**d**) Type-A stilling basin.

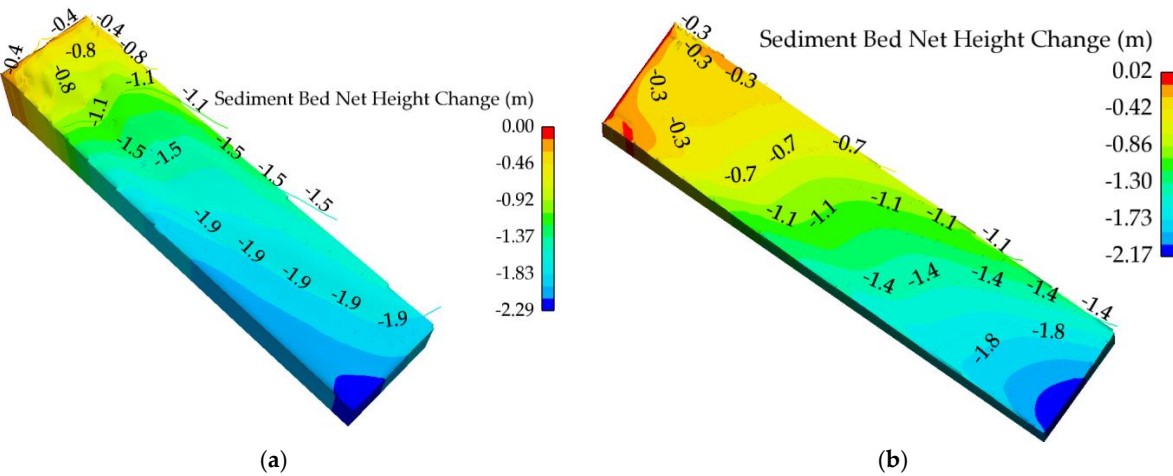

**Figure 11.** *Cont.*

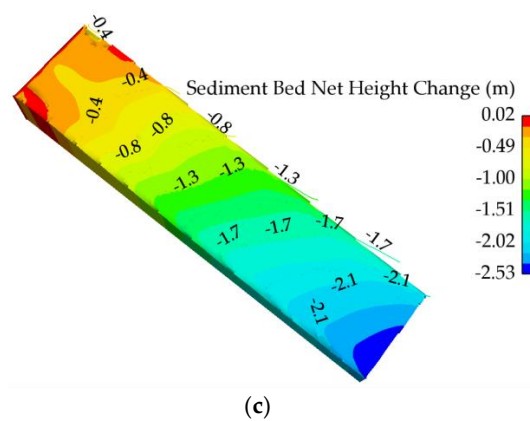

(**c**)

**Figure 11.** Three-dimensional illustration of bed profiles in Type-A Stilling basin using the (**a**) LES turbulence scheme, (**b**) Standard K-ε model, and (**c**) RNG K-ε model.

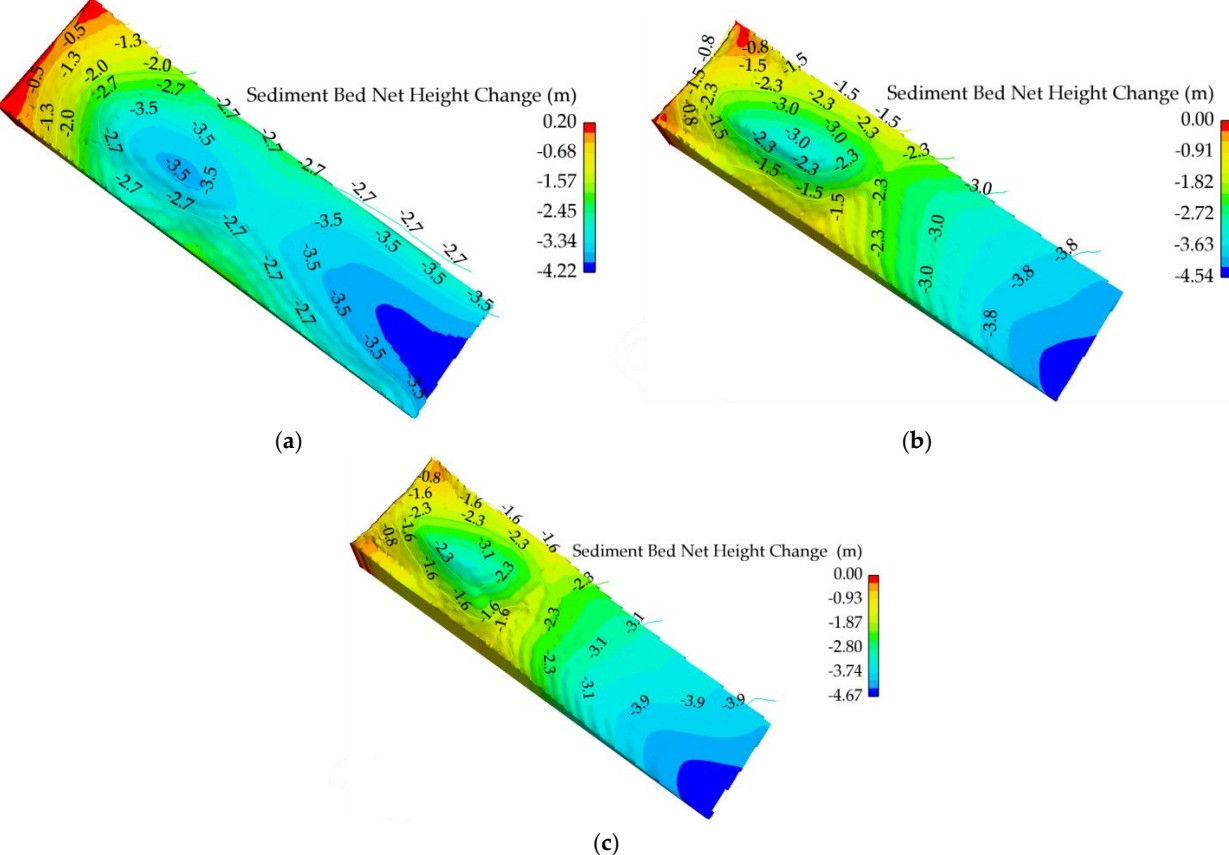

(**a**)

(**b**)

(**c**)

**Figure 12.** Three-dimensional illustration of bed profiles in Type-B Stilling basin using the (**a**) LES turbulence scheme, (**b**) Standard K-ε model, and (**c**) RNG K-ε model.

Figure 12b also showed that after the scour hole, ripples were formed downstream to the end of the scoured bed, and their geometry was found to be dissimilar to that noticed in the LES model. Figure 12c represents the scour pattern in the Type-B stilling basin using the RNG K-ε model, which showed a similar pattern of scour holes as noticed in the Standard K-ε model. The maximum depth of the scour hole and retrogressed bed at the end of the sediment bed was reached at 2.87 m and 4.53 m, respectively. The net height change in the scour hole and end of the sediment reached 60% and 95%, respectively.

### 3.4. Lateral Scour Profiles

Figure 13 shows the lateral profiles of the sediment bed downstream of Type-A and Type-B stilling basins using different turbulence models. The profiles are drawn at various horizontal sections (Y-Z plane), i.e., X = 5, 10, 20, and 30 m, from the end of the rigid bed. Figure 13a shows the bed profile using the LES turbulence model, and results showed that at X = 5 m, maximum scour was found in the centerline and on the left side of the bay, for which scour values reached 0.89 m and 0.78 m, respectively. However, at X = 10, 20, and 30 m, the rate of scour increased on the right side of the bay, reaching 1.51, 2.06, and 2.29 m, respectively. The reason for the higher scour on the right side of the bay was the larger wake area and low-velocity zone in those zones. Figure 13b illustrates bed change in the lateral direction (Y-Z plane) downstream of the Type-A stilling basin using the Standard K-ε model. At X = 5 m, a change in the lateral bed was found, while at X = 10 m, an abrupt change in the bed level was noticed on the right side, for which the net change reached 0.89 m. At X = 20 m, a greater change in the bed elevation was found than in the previous sections, while at X = 30 m, a completely different bed profile was noticed, which indicated maximum bed change in the central region. At X = 30 m, the maximum change in bed was 1.66 m. Figure 13c indicates the lateral bed profile downstream of the Type-A stilling basin using the RNG K-ε model. At X = 5 m, the bed profiles were found to be similar to those obtained in the Standard K-ε model; however, the overall change was less. In the RNG K-ε model, at X = 30 m, the maximum change in bed elevation was reached at 2.53 m.

Figure 13d indicates the lateral change in bed downstream of the Type-B stilling basin using the LES model. At X = 5 m from the rigid bed, the central region of the sediment bed was more scoured than the sides, and a similar pattern was noticed up to X = 20 m. At X = 5, 10, and 20 m, the maximum scour was 2.23, 3.45, and 3.45 m, respectively. The shape and geometries at these sections were also found to be identical; however, at X = 30 m, comparatively flat profiles were obtained. The maximum change in bed was noticed at X = 30 m which reached 4.03 m. Figure 13e shows the lateral bed profiles downstream of the Type-B stilling basin using the Standard K-ε model. Due to the development of the scour hole, at X = 5 m, a net change was found up to 2.69 m. As compared to X = 5 m, the change in bed level at X = 10 m was less; however, a higher change was noticed on the left side due to the larger wake area. The results further showed that at X = 20 and 30 m, the net change in bed was even, reaching 3.07 and 3.92 m, respectively. Using the RNG K-ε model, the overall change in the lateral bed profiles was higher than in the LES and Standard K-ε models, as shown in Figure 13f. The geometry and shape of the scour hole were found to be identical, as was noticed in the Standard K-ε model; however, the depth of the scour hole reached 2.87 m and was found to be less than in other models. Moreover, in the RNG K-ε model, the maximum net change in the lateral bed profile reached 4.67 m at X = 30 m.

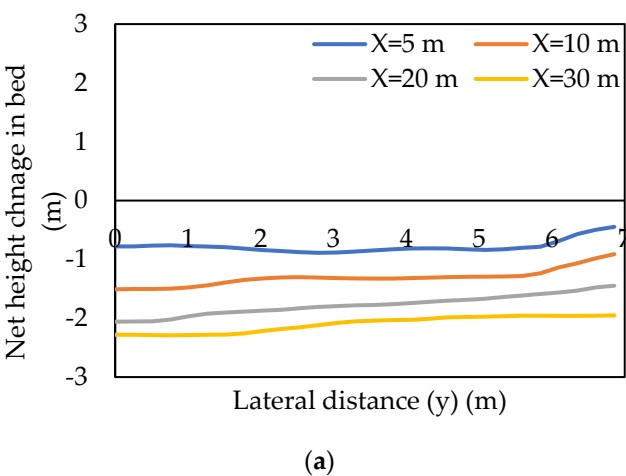

(a)

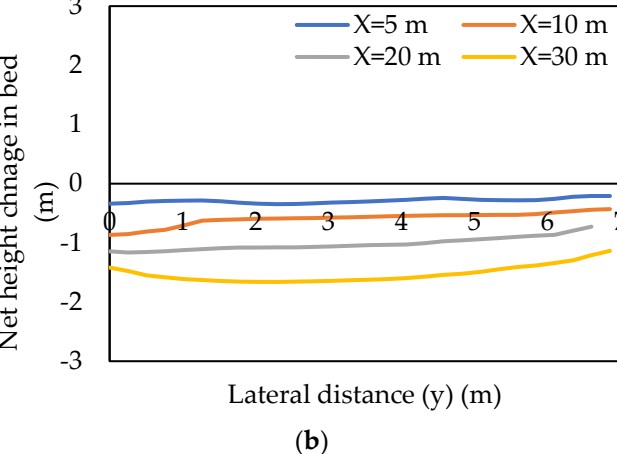

(b)

**Figure 13.** *Cont.*

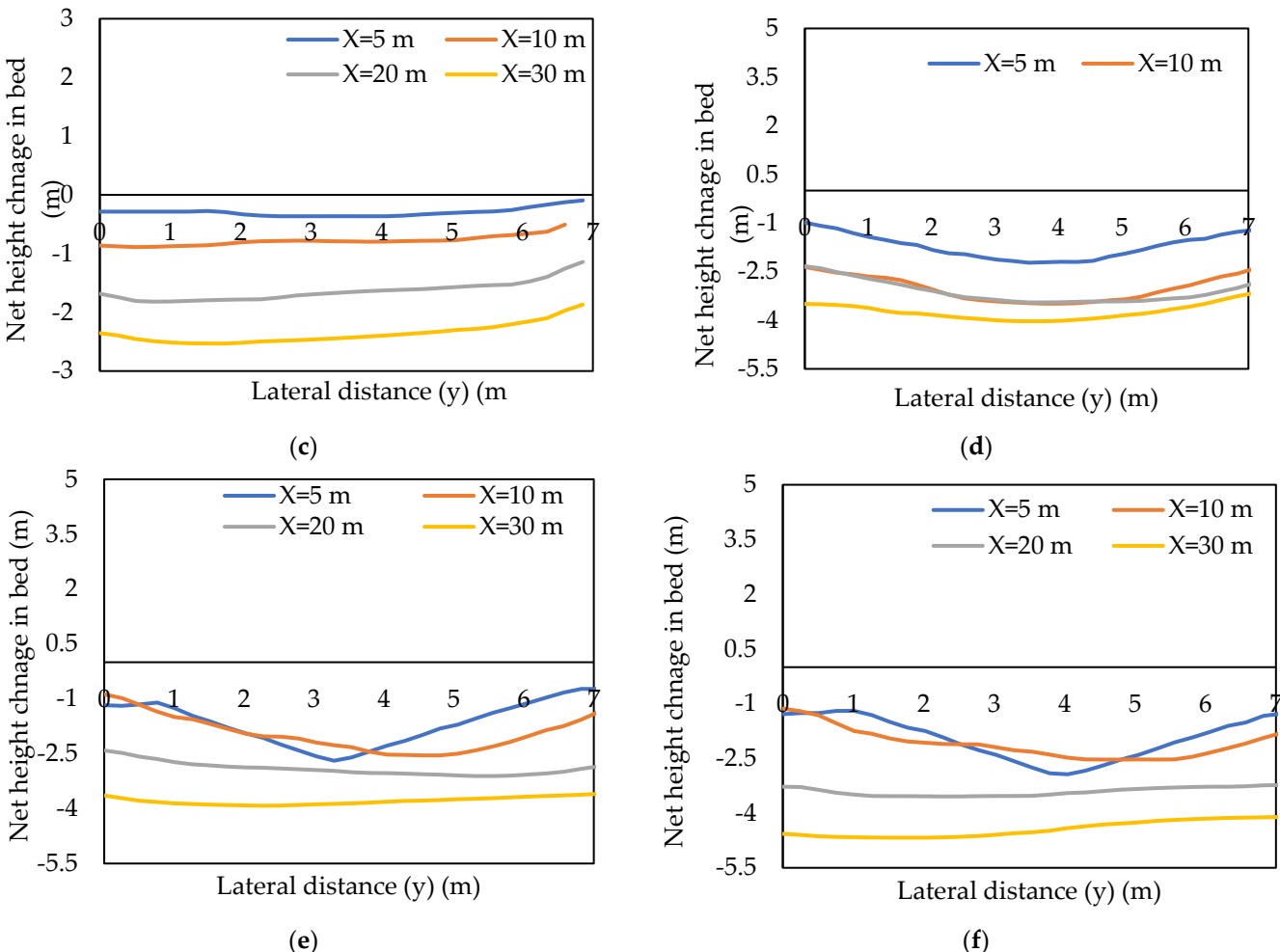

**Figure 13.** Lateral bed profiles in Type-A basin (LES (**a**), Standard K-ε model (**b**), and RNG K-ε model (**c**)), and in Type-B basin (LES (**d**), Standard K-ε model (**e**), and RNG K-ε model (**f**)).

Out of the implemented turbulence models, in both the investigated basins, the RNG K-ε model showed a higher net change in bed elevations. At the initial sections, as compared to the LES model, the net change in the bed elevation was less, while at X = 30 m, a higher change was observed. The minimum bed change in the lateral direction downstream of Type-A and B stilling basins was noticed in Standard K-ε and LES models, respectively.

## 4. Discussion

In the Section 3, different aspects of the present study, such as model performance and validation, flow fields, scour, bed retrogression, and lateral bed profiles, are highlighted in detail. However, the Section 4 only focuses on models' stability, flow fields on the scour bed, maximum scour depth, and scour patterns under different turbulence models.

FLOW-3D scour models were developed to assess the performance of two different stilling basins of the Tuansa barrage at flood discharge. Among the different turbulence models, LES models showed hydraulic stability in both the investigated basins; however, in the Type-B basin, the flow stability was found to be achieved earlier than in the Type-A basin. On the other hand, upon use of Standard K-ε and RNG K-ε models, the present models did not achieve hydraulic stability in the Type-A basin. It is worth mentioning here that downstream of the prototype barrage, similar hydraulic conditions usually occur at such high flows, which show large fluctuations in fluid kinetic energy. Furthermore, due to the difference in the formulation of turbulence models downstream of Type-A and Type-B basins, different values of MAFKE were noticed in different turbulence models. On the contrary, downstream of Type-A and B basins, the analysis showed that scour did

not reach the stability condition. In the field, on UCF conditions (i.e., 24.28 m$^3$/s/m) [41], similar behavior of free surface profiles and scour is generally noticed downstream of the barrage, in which, the riverbed continues to be retrogressed and sometimes large scour pits are observed.

After comparing the maximum scour depths with the field data of bay 33, the results indicated overestimation of the scour depths, which reached to 20%, 23%, and 26% using the LES, Standard K-ε and RNG K-ε models, respectively. For bay 34, the overall errors between the observed and predicted scour depths were reduced to 12%, 9%, and 4% in the LES, Standard K-ε and RNG K-ε models, respectively. On the contrary, after comparing the bed profile with bay 55, the present model underestimates maximum scour depths. However, as compared to LES and standard K-ε models, the results of the RNG K-ε model for bay 55 were close to the observed data, which showed only a 10% error in the maximum scour depth.

After employing the LES model downstream of the Type-B basin, the results of the flow field indicated higher velocity contours from the central fluid depth to the free surface; however, low velocity vectors were seen near the bed, which indicated slow-moving zones of the flow. Using the LES model downstream of the Type-A basin, results showed a non-uniform distribution of the velocity vectors. Like Type-B basins, as compared to the middle and upper fluid layers, lower velocity contours were noticed near the bed, which caused erosion. In the Type-A stilling basin, the LES model showed only forward velocity vectors, which showed bed retrogression. Overall, in both the stilling basins, a non-uniform distribution of the velocity contours was observed from the bed to the free surface. On comparison with the Type-B stilling basin, the velocity values downstream of the Type-A stilling basin were found to be lower. The analysis of the flow field also indicated a wavy free surface downstream of the Type-A basin, which created unsteadiness in the flow streamlines and, as a result, higher bed retrogression. It is important to mention that the erosion of the sediment bed was found because of the slow-moving flow near the bed, and the flow field showed no fluid recirculation on the sediment bed. In general, out of the implemented turbulence models, the results of the flow field using the RNG K-ε model showed higher scour depth and bed retrogression downstream of the Type-B stilling basin. On the contrary, downstream of the Type-A stilling basin, no scour hole was noticed on the entire length of the sediment bed; however, the RNG K-ε model showed a higher net change in the sediment bed.

After analyzing the scour profile, it was revealed that out of the employed turbulence models, the RNG K-ε model showed maximum bed retrogression downstream of the Type-A basin. However, the shape and geometry of ripples and sand waves were found to be dissimilar in different turbulence models. On the other hand, in all the investigated turbulence models, the analysis showed a large scour hole downstream of the Type-B basin. The trend of scour profiles downstream of the Type-B basin was also found to agree with the scour studies [44,45]. After the scour hole, sand ripples were formed, which continued to the end of the scoured bed, and like flow fields, their geometry was also dissimilar in different turbulence models. However, as compared to the LES and Standard K-ε models, the overall length and width of the scour hole developed by the RNG K-ε model was found to be less. Upon use of the RNG K-ε model, the side of the scour hole was also less eroded than in the other turbulence models. Conclusively, for both the investigated basins, the results of longitudinal scour profiles showed dissimilar scouring patterns under different turbulence models. The maximum net change in the sediment beds downstream of Type-A and B stilling basins was noticed in the RNG K-ε model, while the minimum change in bed height was observed in the LES turbulence model.

## 5. Conclusions

The present numerical study investigated local scouring downstream of a river diversion barrage before and after its remodeling using FLOW-3D. Three different turbulence methods were employed to examine the scour patterns and bed retrogression at

24.28 m$^3$/s/m discharge. The modeled scour profiles downstream of the remodeled basin (Type-B) were compared with the probing data of the prototype barrage. The following conclusions are drawn from the present study.

- After comparing scour profiles with the probing data of the prototype, the present models showed overestimation of the scour profiles. However, out of the implemented turbulence models, the scour profiles using the RNG K-ε model showed agreement with the prototype data;
- Using the LES model, irregular velocity fields were observed on the retrogressed and scoured beds of Type-A and Type-B basins, respectively. Upon use of Standard and RNG K-ε models, regular forward velocity fields were noticed on scoured/retrogressed beds, while a backward velocity field was observed within the scour hole downstream of the Type-B basin;
- Under different turbulence schemes, the results of flow fields indicated wake and recirculation zones on the sediment bed near the rigid floor of the Type-B basin, which developed large scour holes. However, downstream of the Type-A basin, slow moving flow near the sediment bed was found to be the main reason for the bed retrogression;
- After analyzing different scour profiles as compared to the LES and Standard K-ε models, the RNG K-ε model indicated higher bed scour/retrogression downstream of the investigated basins. Additionally, as compared to the LES model, the scour patterns and bed retrogression in the Standard and RNG K-ε models were different, showing ripples and a wavy surface. Overall, the net height change in the bed downstream of the Type-B basin reached 95%, while only 51% of the bed was found to be retrogressed downstream of the Type-A basin.

In conclusion, based on the results, it can be said that, as compared to the old basins, the remodeled stilling basin is dissipating less energy and thereby causing higher bed scour. It is further concluded that FLOW-3D is an effective tool for predicting the local scour downstream of the river diversion barrages since the validation results were also found to be promising with the field observation. Furthermore, the results also highlight that both the Van Rijn equation as a bed load transport rate equation and the RNG K-ε turbulence model are efficient for scouring. However, as the present study has investigated a single discharge value and employed only the Van Rijn transport rate equation for the scour, it is recommended to investigate the scour downstream of Taunsa barrage by implementing other discharge values and bed load transport rate equations. The study further suggests employing multiple bays of barrage to investigate the effects of gate openings on scour downstream of the barrage.

**Author Contributions:** Conceptualization, M.W.Z. and I.H.; methodology, M.W.Z., I.H. and S.J.; software i.e., FLOW-3D model, M.W.Z. and S.J.; validation, M.W.Z., S.J. and Z.U.; formal analysis, M.W.Z., I.H. and U.L.; investigation, M.W.Z.; resources, M.W.Z. and I.H.; data curation, M.W.Z.; writing—original draft preparation, M.W.Z.; writing—review and editing, M.W.Z. and I.H.; visualization, M.W.Z. and S.J.; supervision, I.H.; project administration, I.H. All authors have read and agreed to the published version of the manuscript.

**Funding:** This research received no external funding.

**Institutional Review Board Statement:** Not applicable.

**Informed Consent Statement:** Not applicable.

**Data Availability Statement:** All relevant data are included in the paper.

**Acknowledgments:** The authors acknowledge the support of Zulfiqar Ali, Expert Hydraulic for providing prototype data.

**Conflicts of Interest:** The authors declare no conflict of interest.

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
