# Peer review of "Numerical Investigation of Scour Downstream of Diversion Barrage for Different Stilling Basins at Flood Discharge"

_sustainability, doi:10.3390/su151411032_

Round 1

Reviewer 1 Report

Dear Editor,

The  manuscript "Numerical investigation of scour downstream of diversion barrage for different stilling basins at Flood discharge"  is interesting.  

The introduction is rather long. What is this manuscript novelty?

The quality of Fig1 is poor.  Why are some stream patterns not connected to the main pattern?

Fig2a should be improved on the right side, and the scale should also be improved.

The total amount of the figures is weak. 

To better distinguish the direction of velocity, use arrows that illustrate the direction in Figures 9 and 10.

Fig 12 . The quality is poor. 

The manuscript is excessively long.

Reviewer 2 Report

Overview: This manuscript is “Numerical Investigation of Scour Downstream of Diversion Barrage for Different Stilling Basins at Flood Discharge” (sustainability-2412318-peer-review-v1). Some detailed comments are as follows:

Specific comments:

-  Innovative content needs to be added to the abstract of the manuscript; There was a lack of clear research questions or hypotheses in the abstract section of the manuscript; Check the English writing.

- Introduction: the authors check the citations as the guideline for the author; A comprehensive literature review is a prerequisite, and the authors should summarize the research gaps and tell the readers why the authors conduct this study; The study objectives should be shown clearly; Section 1.1 should move to sub-section in Materials and Methods.

-  Materials and Methods: The authors add the flow chart of the study structure.

- The “3. Results and Discussion”, it is recommended to place all the content discussed in a separate section.

-  The research conclusions are more about the research results, and the author needs to further refine them.

-   Proofreading by a native English speaker should be carefully conducted to improve both language and organization quality.

- Proofreading by a native English speaker should be carefully conducted to improve both language and organization quality.

Reviewer 3 Report

The author(s) conducted a numerical investigation of scour downstream of diversion barrage for different stilling basins at flood discharge. The Taunsa barrage in the north Pakistan is considered as the case study. They developed FLOW-3D scour models for this purpose. Overall, it is a good study. The paper is also written well. I have only few minor comments.

1. the value of R2 reached to 0.736, 0.823 and 0.747 for bays 33, 34 and 55. Are these R2 sufficiently high to provide the model performance. Can you please justify.

2. Mention some quantitative results in the abstract.

3. Please add some limitations of the study at the end of the conclusion section

Reviewer 4 Report

The manuscript studied the scour downstream of diversion barrage for different stilling basins at flood discharge, which has certain practical significance and social value. However, the manuscript is too long and lacks a discussion section. Its suggested that the Results Scetion be condensed and that the Discussion Section be expanded.

 The suggested revisions to the manuscript are listed below:

L31, Introduction Section is too long, the space should be compressed

L165, Section 1.1 should be treated as a separate chapter rather than included in the Introduction Section

L293 In Section 2.3, the method and calculation results are mixed together, the two should be strictly distinguished, and the calculation results should be placed in Chapter 3

L411 Explain the source of Figure 7

Round 2

Reviewer 2 Report

The reviewer sees the efforts in editing and perfecting the manuscript by the group of authors. The reviewer's proposal for the manuscript was accepted for publication in the Sustainability Journal.

Reviewer 4 Report

Thanks to the author for spending a lot of time and effort carefully revising the paper according to the suggestions.